# Kansa Method for Unsteady Heat Flow in Nonhomogenous Material with a New Proposal of Finding the Good Value of RBF’s Shape Parameter

**DOI:** 10.3390/ma14154178

**Published:** 2021-07-27

**Authors:** Olaf Popczyk, Grzegorz Dziatkiewicz

**Affiliations:** Department of Computational Mechanics and Engineering, Silesian University of Technology, Konarskiego 18a, 44-100 Gliwice, Poland; Grzegorz.Dziatkiewicz@polsl.pl

**Keywords:** radial basis functions, Kansa method, shape parameter, multiquadric collocation method, inhomogeneous material

## Abstract

New engineering materials exhibit a complex internal structure that determines their properties. For thermal metamaterials, it is essential to shape their thermophysical parameters’ spatial variability to ensure unique properties of heat flux control. Modeling heterogeneous materials such as thermal metamaterials is a current research problem, and meshless methods are currently quite popular for simulation. The main problem when using new modeling methods is the selection of their optimal parameters. The Kansa method is currently a well-established method of solving problems described by partial differential equations. However, one unsolved problem associated with this method that hinders its popularization is choosing the optimal shape parameter value of the radial basis functions. The algorithm proposed by Fasshauer and Zhang is, as of today, one of the most popular and the best-established algorithms for finding a good shape parameter value for the Kansa method. However, it turns out that it is not suitable for all classes of computational problems, e.g., for modeling the 1D heat conduction in non-homogeneous materials, as in the present paper. The work proposes two new algorithms for finding a good shape parameter value, one based on the analysis of the condition number of the matrix obtained by performing specific operations on interpolation matrix and the other being a modification of the Fasshauer algorithm. According to the error measures used in work, the proposed algorithms for the considered class of problem provide shape parameter values that lead to better results than the classic Fasshauer algorithm.

## 1. Introduction

The heat equation is one of the most fundamental parabolic partial differential equations. Therefore, it is often solved in both the world of science and industry. However, it is usually solved with the assumption of spatially constant thermophysical parameters; this is probably related to the fact that thermophysical parameters of naturally occurring materials are usually spatially variable to a small extent, which means that they can be modeled as spatially constant. Nevertheless, in the recent decades, thermal metamaterials, which may exhibit spatial variability of thermophysical parameters, have gained considerable popularity in the scientific community [1,2,3,4]. Particularly noteworthy is the possibility of creating a thermal cloak using a material that directs the heat flow [5]. Due to this, modeling and solving heat flow in metamaterials with spatially variable thermophysical parameters is becoming an important component of computational mechanics. Numerical solving of the heat equation is a well-known issue within classical numerical methods, such as the finite difference method [6] or the finite element method [7]. However, for complicated 2D or 3D geometries of engineering problems, the grid generation process is complex and time-consuming—it may take more time than solving equations that arise for the considered method [8]. Meshless methods, especially the collocation methods, allow to reduce and even avoid such problems. The considered domain is filled with unconnected points for such methods, not a mesh, as in the finite element method. One of the meshless methods is the Kansa method which utilizes the radial basis functions (RBF) for solving partial differential equations. Of course, thermal metamaterials do not exhaust the spectrum of applications of Kansa’s method in the numerical modeling of metamaterials. It can be used, for example, to study radiofrequency metamaterials or microwave metamaterials based on the reconfigurable structures [9,10].

In this paper, the pseudospectral formulation of the Kansa method was used to solve the one-dimensional, unsteady initial-boundary value problem of heat conduction in a non-homogeneous material, which exhibits spatial dependence of thermophysical parameters. Particular emphasis was placed on finding a good shape parameter value; for this purpose, two algorithms for finding it were proposed. As can be seen, the problem under consideration is one-dimensional, in turn, it has been written above that the Kansa method shows its most significant advantage over classical methods when the problem is two-dimensional or three-dimensional. However, the algorithms proposed in this work can be used without any problems in two-dimensional or three-dimensional problems; this is due to the fact that the formulation of the Kansa method is almost identical for one-, two- and three-dimensional problems [11].

The Kansa method was introduced in 1990 by E.J. Kansa [12,13]. In the Kansa method, radial basis functions (RBF) approximate the solution of boundary and initial-value problems of partial differential equations (PDEs). The Kansa method with multiquadric radial basis functions can give outstanding accuracy for the solution of PDEs as shown in [14]; moreover, the method is easy to use for a broad range of models considered in science and engineering. For example, Dubal et al. (1992) [15] used the multiquadric approximation to solve a three-dimensional elliptic PDE describing the time-evolution of interacting black holes and gravitational wave production. Moridis and Kansa (1994) [16] performed time-integration of a certain initial-value problem with the numerical algorithm for inversion of Laplace transformation; the spatial term has been approximated with an exponentially-convergent grid-free scheme using multiquadrics. Sharan et al. (1997) [17] also have used the multiquadric approximation scheme to solve elliptic PDEs with Dirichlet or Neumann boundary conditions. Despite their excellent results, all previous works are related to intuitively applying the Kansa method to solve boundary or initial-value problems of PDEs, and a formal mathematical proof of the convergence [18] is not included in mentioned papers. This state has changed with the publication [19], where authors gave a convergence proof and error bound of the collocation method with radial basis function approximation, at least for PDEs with constant coefficients. In [19], the analysis is based on the fact that this approach can be considered as a special case of the general Hermite–Birkhoff interpolation problem; later, Zerroukat et al. (1998) [20] proposed the scheme with multiquadrics to solve the heat transfer problem. In [18], the Kansa method was applied to solve linear advection-diffusion equations by using the thin-plate splines with the stability proof. Dong and Cheung (2004) [21] utilized the Kansa method to solve elastic inclusion problems. Chantasiriwan (2006) [22] used the multiquadric collocation method to solve the heat conduction problem with stochastic initial and boundary conditions. Zheng and Li (2008) [23] used the Kansa method to investigate acoustic wave propagation. Chen et al. (2010) [24] made the first attempt to solve the time-fractional diffusion equations using the Kansa method with thin-plate splines. Simonenko et al. (2014) [25] used the RBF method to solve elastostatic problems. Pang et al. (2015) [26] applied the Kansa method to the space-fractional advection-dispersion equations. Dehghan and Shirzad (2015) [27] proposed two numerical methods to solve the elliptic stochastic PDEs in two and three dimensions obtained by Gaussian noises using RBFs collocation and pseudospectral collocation methods. Reutskiy (2016) [28] used a meshless radial basis function method for 2D steady-state heat conduction problems in anisotropic and inhomogeneous media. Fallah et al. (2019) [29] used the Kansa approach for solving seepage problems. Haq and Hussain (2019) [30] utilized the Kansa method to solve time-fractional higher-order partial differential equations with constant and variable coefficients. Jankowska and Karageorghis (2019) [31] used the Kansa approach for the numerical solution of second- and fourth-order nonlinear boundary-value problems in two dimensions. Liu and Chang (2019) [32] performed energy regularization of the MQ-RBF method and used the Kansa method for solving the Cauchy problems of diffusion-convection-reaction equations.

Despite all the advantages of the MQ-RBF Kansa method: the accuracy of calculations, the ease of implementation, the ability to apply the method to solve various classes of engineering problems, this method is associated with an unresolved problem that prevents its application to the challenging real-world problems [33]—the problem of choosing an optimal (or at least good) value of the shape parameter in the RBF. For this reason, the problem of choosing the shape parameter has been the subject of great scientific debate. The first attempts to find a good shape parameter value were made even before the Kansa method was invented, MQ-RBF were used for interpolation of scattered data. Hardy (1971) [34] suggested the use of ε=0.815dx, where dx is the distance between equispaced nodes. On the other hand, Franke (1982) [35] recommended $\varepsilon =1.25\sqrt{2}\;dx$. Carlson and Foley (1991) [36] have shown that the shape parameter’s optimal value is strongly dependent on the interpolated function and essentially independent of the number and location of the interpolation nodes. They also gave an algorithm that yields an approximated value of a quasi-optimal shape parameter. Rippa (1999) [37] proposed a leave-one-out cross-validation (LOOCV) algorithm to estimate the interpolation error and use it to compute a quasi-optimal value of the shape parameter; this algorithm was a significant breakthrough; however, it was limited to the problem of interpolation rather than solving PDE using the Kansa method. Fasshauer and Zhang (2007) [38] adapted Rippa’s LOOCV algorithm so that it could be used to find the good value of shape parameter solving PDE problems using the Kansa method together with the multiquadric function. This algorithm is today one of the most popular and reliable algorithms for finding a good value of the shape parameter. Huang et al. (2007) [39] experimentally derived a formula for a good value of the shape parameter $\varepsilon =\log\left(\lambda /3a\;dx\right)$, where *a* and λ are constants that depend on the problem. Bayona (2011, 2012) [40,41] has attempted to find optimal constant and variable shape parameters for the multiquadric approximation method combined with the finite difference method. Tsai et al. (2010) [42] proposed the golden section algorithm, Esmaeilbeigi and Hosseini (2014) [43] the genetic algorithm, Iurlaro et al. (2014) [44] the energy-based approach, and Chen et al. (2010) [14] the sample solution approach. Despite the very intensive development of methods for finding a good value of the shape parameter in the recent years, the choice of the shape parameter for multiply-connected, complex-shaped domain problems remains an open problem [14]. This article is a partial answer to this open problem due to the fact that this publication proposes an algorithm for finding a good value of the shape parameter for a specific class of problems, the reliability of which has been proven experimentally on numerical examples.

## 2. Materials and Methods

### 2.1. Pseudospectral Formulation of Kansa Method for Initial-Value Problem of Heat Conduction

In the classical Fourier model, the heat flux q passing through a material is proportional to the thermal conductivity κ and the local gradient of temperature *u*:(1)q=−κ∇u.

For heat conduction processes without an internal heat source or sink, the equation describing the conservation of internal energy is as follows:(2)$$\gamma \rho u_{t}+\nabla \cdot q=0,$$
where ρ is the density of the medium and γ is the specific heat capacity. After combining Equations (Equation 1) and (Equation 2), the general form of the Fourier equation is obtained
(3)γρut=∇·κ∇u,
which in the one-dimensional case is as follows [45]:(4)γρut=κxux+κuxx.
where κ=κx, ρ=ρx, and γ=γx. Equation (Equation 4) is solved in the time domain t∈0,T and the space domain x,y∈D⊂ℝ2. *T* is the length of the observation interval. Since the Fourier model is parabolic, it is necessary to impose one initial condition for the temperature field:(5)ux,t=0=gx

Three types of boundary conditions were used in the work:Dirichlet boundary condition
(6)$$u\left(x_{b},t\right)=h\left(x_{b}\right),$$
where *h* is boundary temperature function and $x_{b}$ is the coordinate of the boundary point;Neumann boundary condition
(7)$$-\kappa \left(x_{b}\right)u_{n}\left(x_{b},t\right)=q\left(x_{b}\right),$$
where $u_{n}$ is derivative of temperature in the normal direction to the boundary of the considered domain;Robin boundary condition
(8)$$-\kappa \left(x_{b}\right)u_{n}\left(x_{b},t\right)=\alpha \left[u_{\infty }-u\left(x_{b},t\right)\right],$$
where α is heat transfer coefficient and $u_{\infty }$ is a constant ambient temperature.
Equation (Equation 4) together with initial condition and boundary conditions create the initial-boundary value problem which is considered in this work.

The radial basis function is the so-called 2-point real-valued function whose value at a given point *x* depends only on the distance from the selected point $x_{j}$, the so-called center [11]. Every radial basis function satisfies:(9)$$\varphi \left(x,\;x_{j}\right)=\varphi \left(\left|\left|x-x_{j}\right|\right|\right).$$

In this work, the general multiquadric radial basis function was used [33]:(10)$$\varphi \left(x,\;x_{j}\right)={\left({\left(x-x_{j}\right)}^{2}+\varepsilon^{2}\right)}^{p},$$
where ε is a shape parameter. Commonly used values for *p* are −1/2 and 1/2 [33]. In the Kansa collocation method, it is assumed that the unknown field variable, e.g., the field of temperature, at a given point *x* can be expressed as a linear combination of the applied radial basis function [11]:(11)$$u\left(x\right)=\sum_{j=1}^{n}\varphi_{j}\left(x,\;x_{j}\right)c_{j}$$
where $c_{j}$ are coefficients in linear combination and *n* is the number of points in the computational domain. Writing Equation (Equation 11) for all points of the computational domain leads to a n×n system of linear equations, which can be written in a matrix form:(12)u=ϕc,
where ϕ, u, c, are the radial basis functions matrix, temperature field (at collocation points) vector, and coefficients vector, respectively:(13)$$\phi =\left[\begin{array}{cccc} \varphi \left(x_{1}-x_{1}\right) & \varphi \left(x_{1}-x_{2}\right) & \cdots & \varphi \left(x_{1}-x_{n}\right)\\ \varphi \left(x_{2}-x_{1}\right) & \varphi \left(x_{2}-x_{2}\right) & \cdots & \varphi \left(x_{2}-x_{n}\right)\\ \vdots & \vdots & \ddots & \vdots \\ \varphi \left(x_{n}-x_{1}\right) & \varphi \left(x_{n}-x_{2}\right) & \cdots & \varphi \left(x_{n}-x_{n}\right) \end{array}\right],$$
(14)$$u=\left[\begin{array}{c} u_{1}\\ u_{2}\\ \vdots \\ u_{n} \end{array}\right],$$
(15)$$c=\left[\begin{array}{c} c_{1}\\ c_{2}\\ \vdots \\ c_{n} \end{array}\right].$$

Differentiating Equation (Equation 12) with respect to the *x*-coordinate, one can obtain:(16)$$u=\phi c\;\overset{\frac{\partial }{\partial x}}{\to }\;\left\{\begin{array}{c} u_{x}=\phi_{x}c\\ u_{xx}=\phi_{xx}c \end{array}\right.\;\Rightarrow \;\left\{\begin{array}{c} u_{x}=\phi_{x}\phi^{-1}u\\ u_{xx}=\phi_{xx}\phi^{-1}u \end{array}\right.$$
and substitute the obtained u, $u_{x}$, $u_{xx}$ into the heat equation (Equation (Equation 4)) in a matrix form:(17)$$\gamma \rho u_{t}=\left[\kappa_{x}\phi_{x}+\kappa \phi_{xx}\right]\phi^{-1}u.$$

It is assumed that the ϕ matrix is non-singular. The radial basis functions derivatives matrices are as follows:(18)$$\phi_{x}=\left[\begin{array}{cccc} \varphi_{x}\left(x_{1}-x_{1}\right) & \varphi_{x}\left(x_{1}-x_{2}\right) & \cdots & \varphi_{x}\left(x_{1}-x_{n}\right)\\ \varphi_{x}\left(x_{2}-x_{1}\right) & \varphi_{x}\left(x_{2}-x_{2}\right) & \cdots & \varphi_{x}\left(x_{2}-x_{n}\right)\\ \vdots & \vdots & \ddots & \vdots \\ \varphi_{x}\left(x_{n}-x_{1}\right) & \varphi_{x}\left(x_{n}-x_{2}\right) & \cdots & \varphi_{x}\left(x_{n}-x_{n}\right) \end{array}\right],$$
(19)$$\phi_{xx}=\left[\begin{array}{cccc} \varphi_{xx}\left(x_{1}-x_{1}\right) & \varphi_{xx}\left(x_{1}-x_{2}\right) & \cdots & \varphi_{xx}\left(x_{1}-x_{n}\right)\\ \varphi_{xx}\left(x_{2}-x_{1}\right) & \varphi_{xx}\left(x_{2}-x_{2}\right) & \cdots & \varphi_{xx}\left(x_{2}-x_{n}\right)\\ \vdots & \vdots & \ddots & \vdots \\ \varphi_{xx}\left(x_{n}-x_{1}\right) & \varphi_{xx}\left(x_{n}-x_{2}\right) & \cdots & \varphi_{xx}\left(x_{n}-x_{n}\right) \end{array}\right],$$
and the material properties matrices are as follows:(20)$$\kappa =\left[\begin{array}{cccc} \kappa \left(x_{1}\right) & 0 & \cdots & 0\\ 0 & \kappa \left(x_{2}\right) & \cdots & 0\\ \vdots & \vdots & \ddots & \vdots \\ 0 & 0 & \cdots & \kappa \left(x_{n}\right) \end{array}\right],$$
(21)$$\kappa_{x}=\left[\begin{array}{cccc} \kappa_{x}\left(x_{1}\right) & 0 & \cdots & 0\\ 0 & \kappa_{x}\left(x_{2}\right) & \cdots & 0\\ \vdots & \vdots & \ddots & \vdots \\ 0 & 0 & \cdots & \kappa_{x}\left(x_{n}\right) \end{array}\right],$$
(22)$$\gamma =\left[\begin{array}{cccc} \gamma \left(x_{1}\right) & 0 & \cdots & 0\\ 0 & \gamma \left(x_{2}\right) & \cdots & 0\\ \vdots & \vdots & \ddots & \vdots \\ 0 & 0 & \cdots & \gamma \left(x_{n}\right) \end{array}\right],$$
(23)$$\rho =\left[\begin{array}{cccc} \rho \left(x_{1}\right) & 0 & \cdots & 0\\ 0 & \rho \left(x_{2}\right) & \cdots & 0\\ \vdots & \vdots & \ddots & \vdots \\ 0 & 0 & \cdots & \rho \left(x_{n}\right) \end{array}\right].$$

The Euler scheme replaced the time derivative in Equation (Equation 17) [46]:(24)$$u_{t}\approx \frac{u_{k+1}-u_{k}}{\Delta t},$$
which leads to
(25)$$\gamma \rho \frac{u_{k+1}-u_{k}}{\Delta t}=\left[\kappa_{x}\phi_{x}+\kappa \phi_{xx}\right]\phi^{-1}u.$$

The right-hand side of Equation (Equation 25) was discretized using the Crank–Nicolson scheme [47]:(26)$$u\approx \left(1-\sigma \right)u_{k+1}+\sigma u_{k}.$$

Naturally, $\sigma \in \langle 0,\;1\rangle$. Substitution of Equation (Equation 26) into Equation (Equation 25) leads to:(27)$$\gamma \rho \frac{u_{k+1}-u_{k}}{\Delta t}=\left[\kappa_{x}\phi_{x}+\kappa \phi_{xx}\right]\phi^{-1}\left[\left(1-\sigma \right)u_{k+1}+\sigma u_{k}\right],$$
and after rearrangement to:(28)$$\begin{array}{cc} \underset{A}{\underset{︸}{\left[\gamma \rho -\Delta t\left(1-\sigma \right)\left[\left[\kappa_{x}\phi_{x}+\kappa \phi_{xx}\right]\phi^{-1}\right]\right]}}u_{k+1}= & \\ & \underset{B}{\underset{︸}{\left[\gamma \rho +\Delta t\sigma \left[\left[\kappa_{x}\phi_{x}+\kappa \phi_{xx}\right]\phi^{-1}\right]\right]}}u_{k}. \end{array}$$

Equation (Equation 28) is the basic equation solved in this work. Matrices *A* and *B* do not change during time-marching; thus, they will be called time-independent. Equation (Equation 28) can be rewritten in a simpler form:(29)$$Au_{k+1}=Bu_{k}.$$

Solving the transient problem boils down to calculating the consecutive values of $u_{k+1}$ based on the values of $u_{k}$ from the previous time step using classic time-marching:(30)$$u_{k+1}=A^{-1}Bu_{k}.$$

In Equation (Equation 28), boundary conditions are taken into account using the following expressions:$Iu=u_{b}$ for the Dirichlet boundary condition;$\phi_{x}\phi^{-1}u=\pm \kappa^{-1}q$ for the Neumann boundary condition;$\left[\pm \kappa \phi_{x}\phi^{-1}+\alpha \right]u=\alpha u_{\infty }$ for the Robin boundary condition.
I is the identity matrix, $u_{b}$ is the boundary temperature vector, q is the boundary flux vector, α is the heat transfer coefficient diagonal matrix, and $u_{\infty }$ is the ambient temperature vector:(31)$$u_{b}=\left[\begin{array}{c} u_{b1}\\ 0\\ \vdots \\ u_{bn} \end{array}\right],$$
(32)$$q=\left[\begin{array}{c} q_{1}\\ 0\\ \vdots \\ q_{n} \end{array}\right],$$
(33)$$\alpha =\left[\begin{array}{cccc} \alpha \left(x_{1}\right) & 0 & \cdots & 0\\ 0 & 0 & \cdots & 0\\ \vdots & \vdots & \ddots & \vdots \\ 0 & 0 & \cdots & \alpha \left(x_{n}\right) \end{array}\right],$$
(34)$$u_{\infty }=\left[\begin{array}{c} u_{\infty 1}\\ 0\\ \vdots \\ u_{\infty n} \end{array}\right].$$
± shows that the left and right boundary points have different unit normal vectors. Vectors (Equation 31), (Equation 32), and (Equation 34) could be written as 2×1 vectors because in the one-dimensional case, the boundary has only two collocation points; however, the n×1 size was kept to keep the indexation of all matrices consistent.

### 2.2. The Algorithms for Finding the Good Value of Shape Parameter

The multiquadric shape parameter ε is very important for the Kansa collocation method in the presented formulation. Its value significantly influences the condition number of the interpolation matrix and, therefore, the method’s numerical stability and accuracy. In linear algebra, the condition number is the ratio of the largest singular value of a matrix to the smallest one [48]. In the Kansa method application context, recommendations say [39,49,50] that one should use large values of the shape parameter to obtain more accurate solutions. On the other hand, the shape parameter value cannot be too large because the condition number of the interpolation matrix also becomes large, and the accuracy may be lost. Ill-conditioned matrix means a matrix whose condition number is so high that it leads to an unstable simulation. Despite the knowledge of this fact, no algorithm based on calculating the condition number of the interpolation matrix has been proposed so far; it probably results from the fact that the operation of calculating the condition number is computationally expensive. However, the increasing computing power of computers makes this problem less and less significant. Consequently, algorithms for searching for a good shape parameter based on this numerical operation can potentially be used on a large scale. In the present work, two algorithms for finding a good value of the shape parameter for the problem of one-dimensional, unsteady heat flow in a material with spatially variable thermophysical parameters were proposed.

The first algorithm proposed in this paper, called the condition algorithm (CA), uses the condition number as an indicator of the system behavior; using Equation (Equation 28) it could be defined as follows:(35)$$L\left(\varepsilon \right)=\mathrm{cond}\left(A^{-1}B\right).$$

It should be emphasized that the typical shape of the function $L\left(\varepsilon \right)$ for the considered class of problems is shown in Figure 1. It is possible to find a good value of the shape parameter $\varepsilon_{\mathrm{CA}}$ using the graph of the condition number. It is assumed that a good value of the shape parameter indicates the start of the oscillatory behavior of the $L\left(\varepsilon \right)$ function, as shown in Figure 1. The fundamental question is how to detect the onset of an oscillation. It was proposed to study changes in the monotonicity of the function $L\left(\varepsilon \right)$. If there are two monotonicity changes in a row—it is considered that oscillations begin there. Algorithm 1 presents a pseudocode describing the proposed approach.
**Algorithm 1.** Pseudocode describing the proposed condition algorithm.
X is the vector of collocation points coordinates, ε is a vector of the shape parameter values. The range of ε was selected empirically by trial and error.
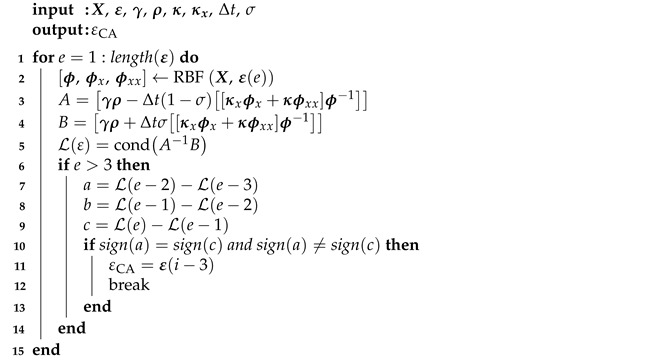


The second algorithm, called modified Fasshauer’s algorithm (MFA) combines Fasshauer’s algorithm (FA) [38] with the approach presented in the condition algorithm. In the Fasshauer approach, a minimum of the cost function is sought [38], while in the proposed MFA, ε indicating the oscillatory regime of the graph—similar to the condition algorithm. Figure 2 shows a comparison of approaches in these two algorithms. Algorithm 2 presents a pseudocode describing the proposed approach.
**Algorithm 2.** Pseudocode describing the proposed modified Fasshauer algorithm.
X is the vector of collocation points coordinates, ε is a vector of the shape parameter values. The range of ε was selected empirically by trial and error.
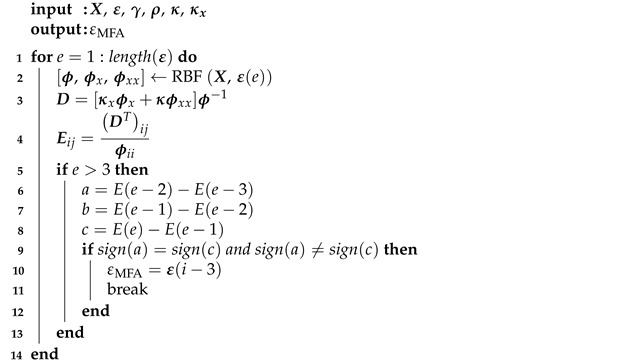


### 2.3. Reference Solutions

To investigate the accuracy of the presented methods, the obtained results were compared with the solutions obtained with two other methods:The analytical method for the steady-state;The finite difference method for the transient analysis.

If in the heat equation (Equation (Equation 4)) it is assumed that the derivative of the temperature with respect to time is equal to zero, then the heat equation is simplified to the steady-state heat equation:$$\kappa_{x}u_{x}+\kappa u_{xx}=0$$
for which a steady-state analytical solution is known for a certain group of functions $\kappa \left(x\right)$. This solution can be compared with the solution from the last time step of the numerical solution.

In this work, the implicit finite-difference method was used to maintain the stability of the simulation in small time steps. Central discretization in space and forward-central discretization in time were used:(36)$$\gamma_{i}\rho_{i}\frac{u_{i}^{k+1}-u_{i}^{k}}{\Delta t}=\kappa_{xi}\frac{u_{i+1}^{k+1}-u_{i-1}^{k+1}}{2\Delta x}+\kappa_{i}\frac{u_{i+1}^{k+1}-2u_{i}^{k+1}+u_{i-1}^{k+1}}{{\left(\Delta x\right)}^{2}}$$

After reorganizing the terms, the equation takes the following form:(37)$$\begin{array}{cc} \left(\frac{\kappa_{xi}}{2\Delta x}-\frac{\kappa_{i}}{{\left(\Delta x\right)}^{2}}\right)u_{i-1}^{k+1}+\left(\frac{\gamma_{i}\rho_{i}}{\Delta t}+\frac{2\kappa_{i}}{{\left(\Delta x\right)}^{2}}\right)u_{i}^{k+1}+ & \\ & \left(-\frac{\kappa_{xi}}{2\Delta x}-\frac{\kappa_{i}}{{\left(\Delta x\right)}^{2}}\right)u_{i+1}^{k+1}=\frac{\gamma_{i}\rho_{i}}{\Delta t}u_{i}^{k} \end{array}$$

The left-hand side contains the unknown values of variables, while the right-hand side contains the known values from the previous time steps. Equation (Equation 37) must be arranged for each point of the domain, and then, the system of equations needs to be solved, taking into account the boundary conditions. Figure 3 shows the stencil of the implicit scheme used.

### 2.4. Error Measures

Three error measures were used in this work:Mean percentage error calculated between the last time step solution of the Kansa method and the analytical steady-state solution
(38)$$\Delta_{AK}=\frac{\int_{0}^{l}u_{A}\left(x\right)dx-\int_{0}^{l}u_{K}\left(x\right)dx}{\int_{0}^{l}u_{A}\left(x\right)dx}\cdot 100\%,$$
where $u_{A}$ is the steady-state analytical temperature field and $u_{K}$ is the Kansa method solution temperature field. This measure was used to investigate whether the Kansa method solution converges to the steady-state analytical solution.Mean percentage error calculated between the Kansa method solution and the finite-difference method solution
(39)$$\Delta_{FK}=\frac{1}{m-1}\sum_{k=2}^{m-1}\frac{\int_{0}^{l}u_{F,k}\left(x\right)dx-\int_{0}^{l}u_{K,k}\left(x\right)dx}{\int_{0}^{l}u_{F,k}\left(x\right)dx}\cdot 100\%,$$
where $u_{F,k}$ is the finite difference method solution temperature field at *k*-th time step while $u_{K,k}$ is the Kansa method solution temperature field at *k*-th time step. This measure was used to investigate whether the convergence of the Kansa method solution to the steady-state is correct. The first time step was not taken into account because the system’s state is then the initial condition.Mean percentage error calculated between the finite-difference method solution and the steady-state analytical solution
(40)$$\Delta_{AF}=\frac{\int_{0}^{l}u_{A}\left(x\right)dx-\int_{0}^{l}u_{F}\left(x\right)dx}{\int_{0}^{l}u_{A}\left(x\right)dx}\cdot 100\%,$$
where $u_{A}$ is the analytical steady-state solution temperature field and $u_{F}$ is the finite-difference method solution temperature field. This measure was used to validate the previous measure by checking if the finite-difference method solution converged to the analytical steady-state solution.

## 3. Numerical Results and Discussion

### 3.1. General Insights about Numerical Setup and Thermal Parameters Distributions

The considered one-dimensional problems were computed in the spatial domain of $x\in \left(0,\;L\right)$. Four distributions of thermophysical parameters were used in the numerical examples: linear, exponential, harmonic and discontinuous. Figure 4 presents visualizations of three of them along with equations describing them.

The values of the coefficients appearing in the distributions shown in Figure 4 are as follows a=1, b=1.2, k=4, L=2m. The constants *a* and *b* have appropriate units depending on the physical quantity represented.

### 3.2. Discontinuous Distributions of Thermal Parameters, the Influence of Sharpness Parameter

It is often necessary to study the heat flow in a material whose thermal parameter distributions are spatially variable but discontinuous. In this situation, one can divide the domain into subdomains in which the distribution of thermal parameters are continuous functions, then solve the problem for each subdomain, and then connect the solutions. Unfortunately, in some cases, the approach is not favorable from an implementation point of view. For this reason, it was decided to use a function that models the jump between two values, a linear combination of such functions would give the desired discontinuous distribution. One of the most popular of these generalized functions is the Heaviside step function:(41)$$H\left(x\right)=\left\{\begin{array}{cc} 0 & \mathrm{if}\;x<0\\ \frac{1}{2} & \mathrm{if}\;x=0\\ 1 & \mathrm{if}\;x>0 \end{array}\right.$$

Unfortunately, the use of the Heaviside step function in its exact form presented in Equation (Equation 41) is quite impractical from the programming point of view because it requires the use of conditional operators. For this reason, it was decided to use its analytical approximation, more specifically, the logistic function [51]:(42)$$H\left(x\right)\approx \frac{1}{1+\exp\left(-2sx\right)},$$
where *s* is the sharpness parameter specifying how abrupt is the transition between two values in the analytical approximation of the Heaviside step function. The impact of the *s* parameter is shown in Figure 5.

The derivative of the logistic function (Equation (Equation 42)) is as follows:(43)$$\frac{\partial H}{\partial x}\approx \frac{2s\;\exp\left(-2sx\right)}{{\left(1+\exp\left(-2sx\right)\right)}^{2}}.$$

The graphical interpretation of the function (Equation 43) is a Gaussian-like (bell-like) curve which has properties similar to the normal distribution curve; the higher the value of the s parameter, the slimmer the bell-like function graph is, as in the normal distribution while decreasing the value of the standard deviation. Figure 6 presents a graphic interpretation of the function (Equation 43) together with the influence of the *s* parameter on it.

The selection of the sharpness parameter *s* generates some problems. At first glance, it may seem that the higher its value, the higher the accuracy of the method because of the discontinuities in the distributions of thermal parameters are reproduced more accurately. This is true, however, the problem lies somewhere else—in the derivative of the logistic function. As already mentioned, its graphical interpretation is the bell-like curve, the consequence of which is that it tends to Dirac’s impulse when the sharpness parameter *s* tends to infinity. This makes it impossible to increase the value of the *s* parameter arbitrarily because, at a sufficiently high value, the curve will be so slender and reach such high values that the solution obtained will be affected by large inaccuracies originating from the nature of numerical calculations and the digital systems arithmetic. To answer the question of how to choose the value of the *s* parameter, its impact on $\Delta_{AK}$, $\Delta_{FK}$, and $\Delta_{AF}$ error measures was examined. The numerical setup was as follows:p=0.5;σ=0;Thermal parameters distribution—discontinuous;Number of collocation points—80, 100, and 120;Boundary conditions type—Dirichlet & Dirichlet;Time step size—1s;Number of time steps—10.

Figure 7 shows the discontinuous distribution of thermal parameters symbolically that has been approximated. Figure 8, Figure 9 and Figure 10 show the influence of the sharpness parameter *s* on $\Delta_{AK}$, $\Delta_{FK}$, and $\Delta_{AF}$ error measures for the number of collocation points equals to 80, 100, and 120, respectively. The value of the shape parameter was selected using the condition algorithm.

Figure 8, Figure 9 and Figure 10 confirm the correctness of the above considerations regarding the selection of the sharpness parameter *s*—it can not be too small because the discontinuities are smeared or too large because the derivatives reach tremendous values. It is worth noting that the optimal value of the sharpness parameter is dependent on the number of collocation points, the denser the mesh, the higher the optimal value of the sharpness parameter *s*—the location of the peak values indicates this in Figure 8, Figure 9 and Figure 10. This is consistent with the predictions, the denser the mesh, the more accurately the abrupt changes can be modeled and, as a consequence, a larger value of sharpness parameter *s* can be used. Nevertheless, it is difficult to indicate the clear rule for choosing this value.

### 3.3. Influence of σ Coefficient

Since the Crank–Nicolson scheme was considered in this paper, an important question arises: what should be the percentage shares of the explicit scheme and the implicit scheme in the Crank–Nicolson scheme to obtain the highest accuracy? In other words, what should be the value of the σ coefficient? To answer this question, the influence of the σ coefficient on $\Delta_{AK}$, $\Delta_{FK}$, and $\Delta_{AF}$ error measures was examined. The simulations were performed in the range of $\langle 0,\;0.6\rangle$, for higher values, the simulation was unstable because the numerical scheme becomes mostly explicit. The numerical setup was as follows:p=0.5;thermal parameters distribution—harmonic;the number of collocation points n=100;applied boundary conditions: Dirichlet on both boundaries;the size of the time step, Δt=0.5s;the number of time steps—20.

The shape parameter was selected using the condition algorithm for each value of σ. Figure 11 shows the influence of the σ coefficient on the $\Delta_{AK}$ and $\Delta_{FK}$ error measures.

The results presented in Figure 11 lead to several interesting conclusions. The function $\Delta_{FK}\left(\sigma \right)$ is monotonically increasing in the whole considered range, while the function $\Delta_{AK}\left(\sigma \right)$ has a minimum. On this basis, the value σ=0 is recommended. There are three reasons for this:The measure $\Delta_{FK}\left(\sigma \right)$ is more important because it describes the quality of the solution in the transient state and not only at the last time step;The difference between $\Delta_{AK}\left(0\right)$ and $\Delta_{AK}\left(\sigma_{\mathrm{MIN}}\right)\;=\min\left(\Delta_{AK}\right)$ is small while the difference between $\Delta_{FK}\left(0\right)$ and $\Delta_{FK}\left(\sigma_{\mathrm{MIN}}\right)$ is significant;The value of $\sigma_{\mathrm{MIN}}$ depends on the numerical setup while σ=0 does not.

### 3.4. The Algorithms for Finding the Good Value of the Shape Parameter

To examine the quality of the proposed algorithms, four test cases were simulated. Table 1 shows their numerical setups. The length of the time steps was selected arbitrarily in such a way as to show the effectiveness of the algorithms for different sizes of the time step. The same approach was used for the *p* parameter. In all cases, σ=0.

Figure 12, Figure 13, Figure 14 and Figure 15 show a summary of good shape parameter values obtained with various algorithms and error norms for simulations carried out with these shape parameter values. Figure 16, Figure 17, Figure 18 and Figure 19 show the graphs of the function $L\left(\varepsilon \right)$ and the cost function with good shape parameter values marked on them determined using the condition algorithm ($\varepsilon_{\mathrm{CA}}$), Fasshauer’s algorithm ($\varepsilon_{\mathrm{FA}}$), and modified Fasshauer’s algorithm ($\varepsilon_{\mathrm{MFA}}$).

The data presented in Figure 12, Figure 13, Figure 14, Figure 15, Figure 16, Figure 17, Figure 18 and Figure 19 lead to several conclusions. In this discussion, the statement that the algorithm is accurate/precise should be understood that the algorithm gives a shape parameter value that gives accurate/precise results from of the error measures presented earlier. In turn, reliability should be understood as the ability of the algorithm to give such a shape parameter value that will most likely not lead to an ill-conditioned interpolation matrix. The most important conclusion is that the proposed condition algorithm is both precise and reliable. In all considered test cases, the values of error norms did not exceed 0.3% when using it. In some cases, the other algorithm gave better results, but the differences were insignificant, for example, in test case 2, the results of which are shown in Figure 13. However, more important is reliability; in none of the considered test cases, the shape parameter value was high enough to make the interpolation matrix ill-conditioned, which indicates a high reliability of the algorithm. This fact is a significant issue because for this class of problems, Fasshauer’s algorithm did not provide reliability; in two cases, the obtained shape parameter value caused that the interpolation matrix was ill-conditioned. It is worth noting that it is difficult to state which numerical setup parameter has the most significant impact on the ill-condition of the interpolation matrix; it is probably the effect of the combined parameters. It should be emphasized once again that the condition algorithm is definitely more computationally expensive than Fasshauer’s algorithm due to the fact that it requires repeated calculation of the conditioning number, which in itself is a costly operation. As for the modified Fasshauer algorithm, the quality of its results is somewhat between the previous two algorithms. It is slightly more reliable than Fasshauer’s algorithm because only for test case 4 it gave an ill-conditioned interpolation matrix. As for the values of error measures, they were in all test cases larger than for the condition algorithm; however, they were usually at a reasonable level not exceeding 0.63%. It is worth noting that both Fasshauer’s algorithm and modified Fasshauer’s algorithm work much better when the cost function has an apparent minimum.

## 4. Conclusions

The discussion carried out in the previous section allowed us to formulate the following conclusions:The condition algorithm proposed in this work is a very reliable and precise algorithm for choosing a good shape parameter value for the considered class of problems. Particularly noteworthy is its ability to give shape parameter values that do not cause the interpolation matrix to be ill-conditioned. It is worth mentioning that the algorithm is more computationally expensive than classic algorithms such as the Fasshauer algorithm.The modified Fasshauer algorithm is an interesting alternative to the classic Fasshauer algorithm. It gives slightly greater reliability than Fasshauer’s algorithm; however, not as good as the condition algorithm. The accuracy of simulation results using it is slightly worse than using Fasshauer’s algorithm and the condition algorithm.

Recommendations based on the preliminary results:The suggested value of σ for the considered class of problems due to the error measures $\Delta_{AK}$ and $\Delta_{FK}$ is 0;The selection of the optimal value of the sharpness parameter *s* is problematic. Based on the performed study, it is not possible to formulate a general selection rule, but some recommendations may be mentioned. The *s* value should be chosen so that the ∂H/∂x does not reach a tremendous value. It is worth noting that the higher the number of collocation points, the higher the optimal value of *s*.

## Figures and Tables

**Figure 1 materials-14-04178-f001:**
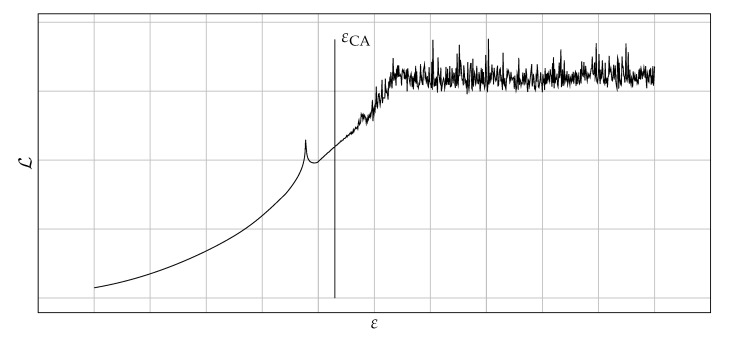
Example of $L\left(\varepsilon \right)$ function with marked good value of the shape parameter ε.

**Figure 2 materials-14-04178-f002:**
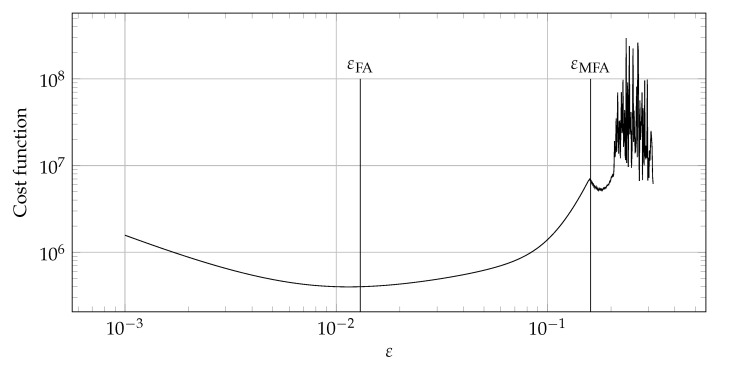
Sample of a graph of the cost function. $\varepsilon_{\mathrm{FA}}$ is equal to the ε for which the cost function reaches minimum while $\varepsilon_{\mathrm{MFA}}$ is equal to the ε for which the oscillatory behavior begins.

**Figure 3 materials-14-04178-f003:**
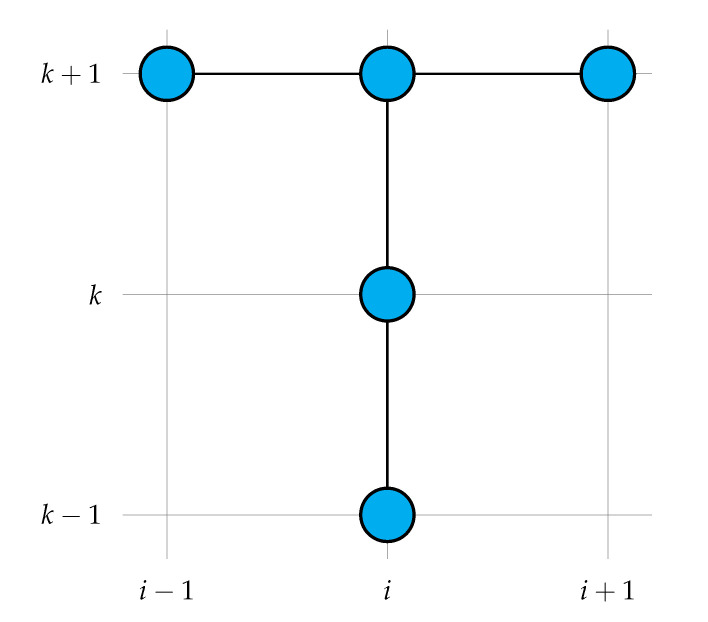
The stencil of the implicit scheme used in this work.

**Figure 4 materials-14-04178-f004:**
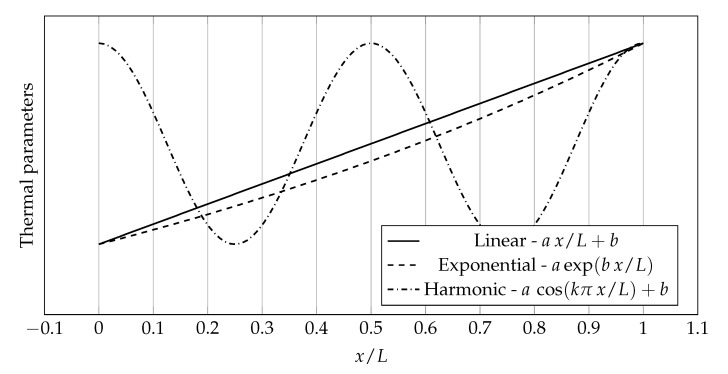
Distribution shapes of thermophysical parameters used in the calculations.

**Figure 5 materials-14-04178-f005:**
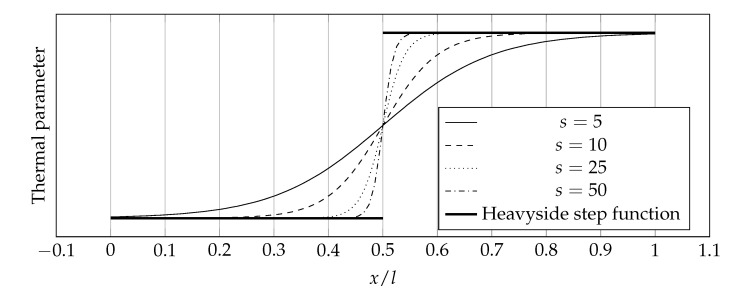
Discontinuous distribution of thermal parameters modeled using an analytical approximation of the Heaviside step function in the form of the logistic function for various values of *s*.

**Figure 6 materials-14-04178-f006:**
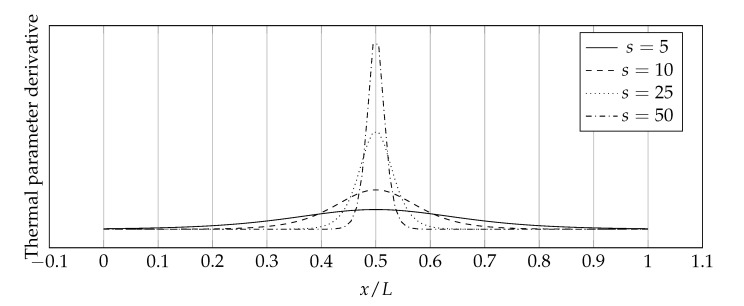
Derivative of the logistic function for various values of *s*.

**Figure 7 materials-14-04178-f007:**
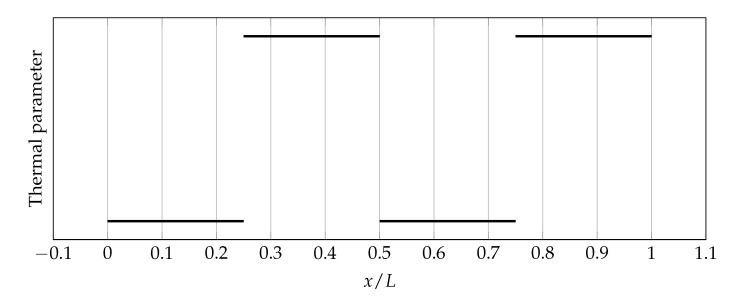
Discontinuous distribution of thermal parameters that has been approximated.

**Figure 8 materials-14-04178-f008:**
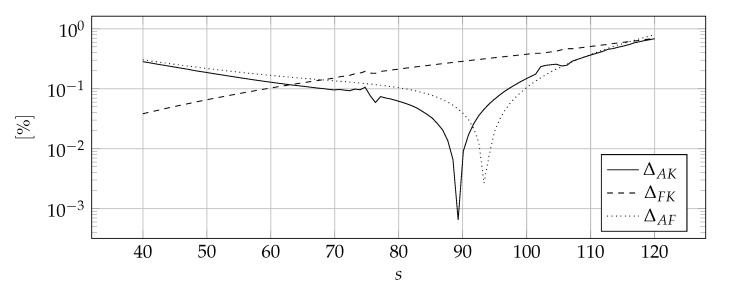
The influence of the sharpness parameter *s* on $\Delta_{AK}$, $\Delta_{FK}$, and $\Delta_{AF}$ error measures for 80 collocation points.

**Figure 9 materials-14-04178-f009:**
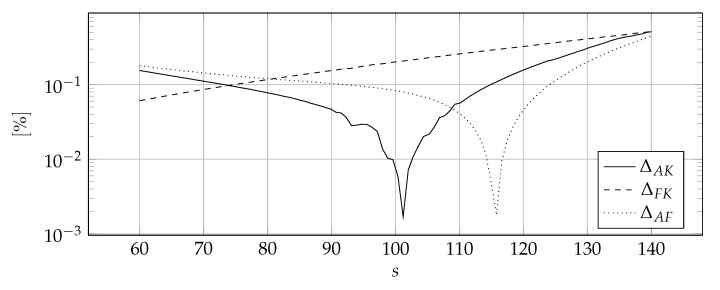
The influence of the sharpness parameter *s* on the $\Delta_{AK}$, $\Delta_{FK}$, and $\Delta_{AF}$ error measures for 100 collocation points.

**Figure 10 materials-14-04178-f010:**
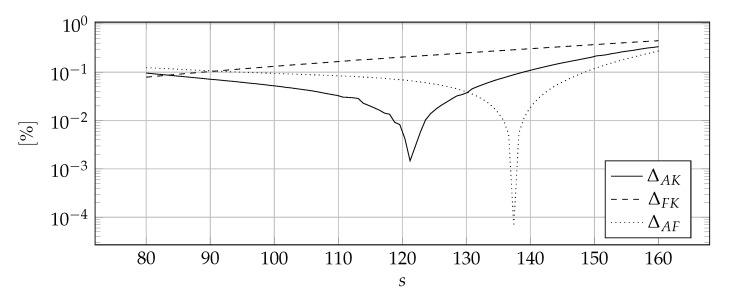
The influence of the sharpness parameter *s* on the $\Delta_{AK}$, $\Delta_{FK}$, and $\Delta_{AF}$ error measures for 120 collocation points.

**Figure 11 materials-14-04178-f011:**
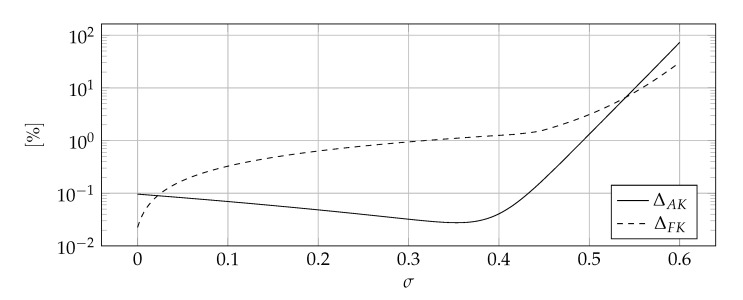
The influence of σ coefficient on $\Delta_{AK}$, $\Delta_{FK}$ error measures.

**Figure 12 materials-14-04178-f012:**
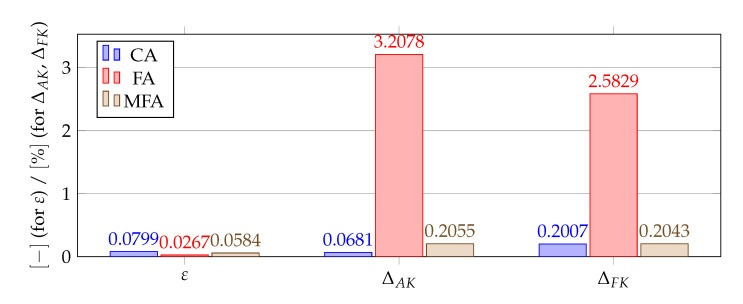
Comparison of shape parameters and error measures obtained using different algorithms for finding a good value of the shape parameter for test case 1, $\Delta_{AF}=0.0674\%$.

**Figure 13 materials-14-04178-f013:**
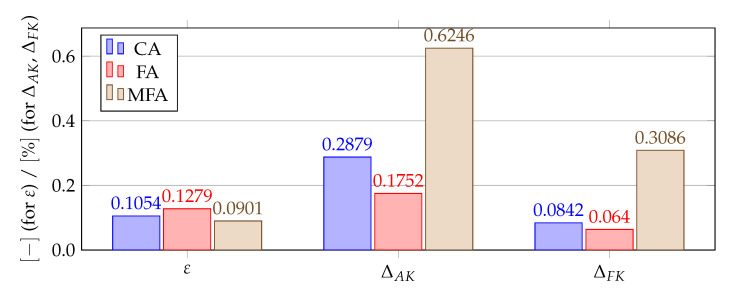
Comparison of shape parameters and error measures obtained using different algorithms for finding a good value of the shape parameter for test case 2, $\Delta_{AF}=0.1469\%$.

**Figure 14 materials-14-04178-f014:**
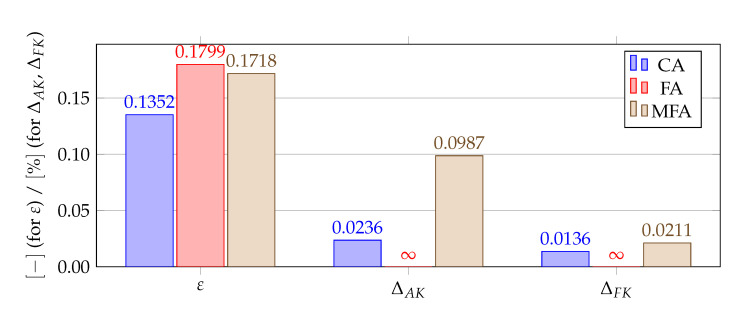
Comparison of shape parameters and error measures obtained using different algorithms for finding a good value of the shape parameter for test case 3, $\Delta_{AF}=0.0082\%$. The value of *∞* means that the given algorithm led to an ill-conditioned interpolation matrix making the simulation unstable.

**Figure 15 materials-14-04178-f015:**
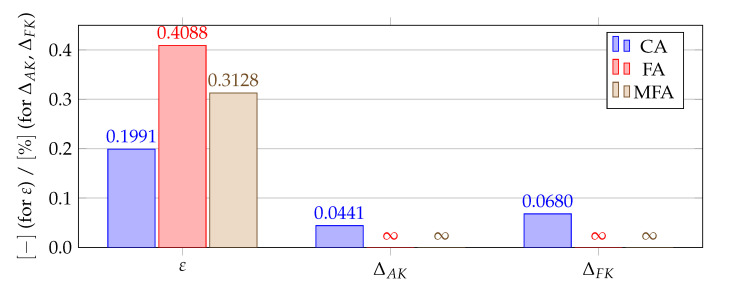
Comparison of shape parameters and error measures obtained using different algorithms for finding a good value of the shape parameter for test case 4, $\Delta_{AF}=0.0224\%$. The value of *∞* means that the given algorithm led to an ill-conditioned interpolation matrix making the simulation unstable.

**Figure 16 materials-14-04178-f016:**
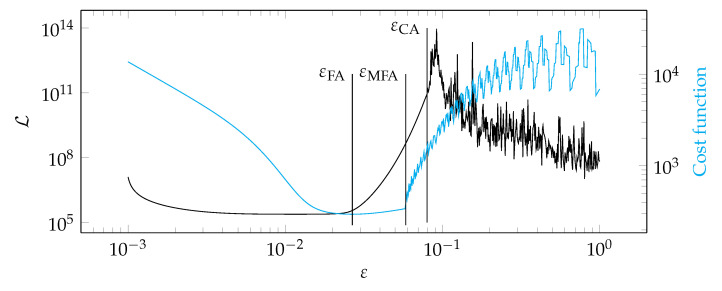
Graphs of the function $L\left(\varepsilon \right)$ and the cost function for test case 1.

**Figure 17 materials-14-04178-f017:**
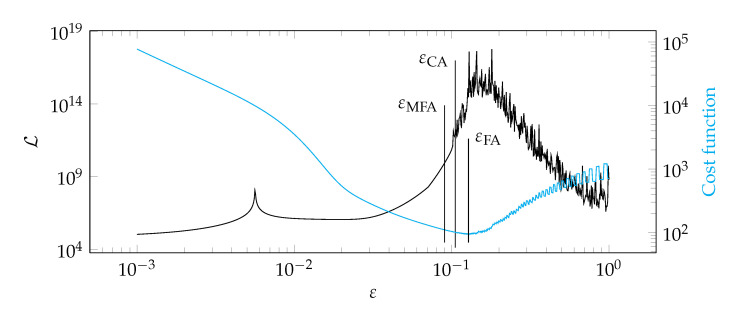
Graphs of the function $L\left(\varepsilon \right)$ and the cost function for test case 2.

**Figure 18 materials-14-04178-f018:**
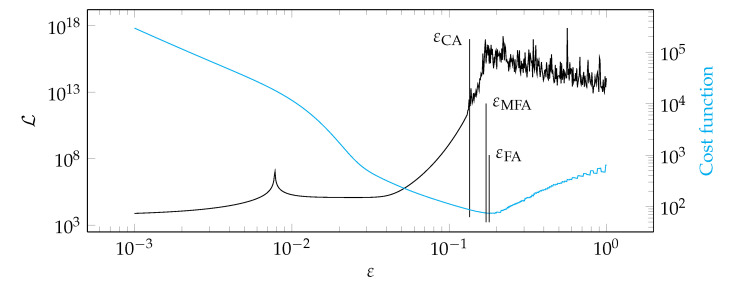
Graphs of the function $L\left(\varepsilon \right)$ and the cost function for test case 3.

**Figure 19 materials-14-04178-f019:**
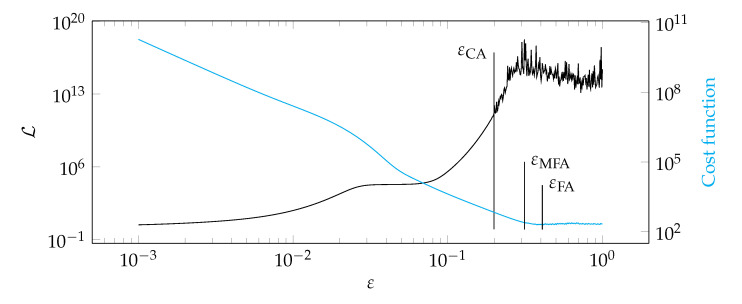
Graphs of the function $L\left(\varepsilon \right)$ and the cost function for test case 4.

**Table 1 materials-14-04178-t001:** Data of numerical examples.

Test Case	1	2	3	4
*p*	0.5	0.4	0.3	−0.5
Parameters distribution	Discontinuous	Harmonic	Exponential	Linear
Collocation points number	240	200	160	120
BC at left edge	Robin	Neumann	Dirichlet	Neumann
BC at right edge	Robin	Robin	Robin	Dirichlet
Time step size	0.1 s	0.5 s	1 s	2 s

## Data Availability

If one want to receive numerical data, please contact the authors.

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
