# Peer review of "Kansa Method for Unsteady Heat Flow in Nonhomogenous Material with a New Proposal of Finding the Good Value of RBF’s Shape Parameter"

_materials, 2021, doi:10.3390/ma14154178_

Round 1
Reviewer 1 Report
This manuscript studied unsteady heat flow in nonhomogenous material with Kansa method, and proposed new algorithms for finding the good value of shape parameter. The following comments should be addressed before acceptation.
- Sections are not well formatted and arranged in the manuscript, and the writing should be much improved. I suggest not using subsections in Introduction section.
- Since this work studies the unsteady heat flow, why did the authors compare the proposed methods to the steady-state method? Please explain that.
Author Response
First, we would like to thank the Reviewers for their contribution. Their suggestions were beneficial during the revision process and have been incorporated into the revised manuscript. They helped us understand the paper's weak points and, as a direct consequence, we believe the quality of the revised paper has been improved significantly. All questions and comments are addressed below. The red fonts indicate the places of changes in the paper.
- Sections are not well formatted and arranged in the manuscript, and the writing should be much improved. I suggest not using subsections in Introduction section.
We improved the writing and arranging of the text. Subsections have been deleted from the Introduction.
- Since this work studies the unsteady heat flow, why did the authors compare the proposed methods to the steady-state method? Please explain that
Generally speaking, the problem considered in the paper does not have known analytical solutions for transient analysis; however, analytical solutions are known in particular cases for steady-state boundary-value problems. The numerical solutions for transient analysis obtained in the presented study were compared with the transient solution, obtained by the finite difference method (FDM) (the error measure given by Eq. (39) indicates the difference between obtained unsteady solutions) and the steady-state analytical solution. The steady-state analytical solution allows examining the quality of the numerical solution for the last time step of the transient analysis close to the asymptotic state, which provides information, for example, on the correctness of implementing the considered meshless method.
Reviewer 2 Report
The paper "Kansa Method for Unsteady Heat Flow in Nonhomogenous Material with a New Proposal of Finding the Good Value of RBF’s Shape Parameter" by Olaf Popczyk and Grzegorz Dziatkiewicz presents the results of application of the Kansa method in the pseudospectral formulation to solve the one-dimensional, unsteady initial-boundary value problem of heat flow in a material with spatially variable thermophysical parameters. The study of the appliacation of two new algorithms for finding a good shape parameter value based on the analysis of the condition number of the interpolation matrix are performed. At the same time a modification of the Fasshauer algorithm are proposed.
I think the manuscript can be published in Materials since the considere method is approproate for material science scope. Before next round for publication Authors should reconsider their text and perform some improvements (listed below) and after the following recommendations are considered by the authors. My remarks are as follows:
1) The Abstact should be focused at the scope of Materias. Now it is very labyrinthine with numerical methods. Authors should rewrite the Abstract to emphasize the main findings of their work and convince the reader that this approach is eligible for Material science.
2) The Introduction focused mainly on the solving of heat flow equations in metamaterials. Since the author cite the Ref[1]-[4] describing the thermal metamaterials, the radiofrequency metamaterials should also be mentioned as a good candidate for the Kansa method of numerical simulation. Thus, it will be more instructive if
[Journal of Communications Technology and Electronics, 2014, Vol. 59, No. 9, pp. 914–919], [JETP Letters, 2018, Vol. 107, No. 1, pp. 25–29] are contained in the reference list along with Ref. 1-4 pointing that the Microwave metamaterials based on the reconfigurable structures can also be candidate to the beforementioned approach.
3) The Authors should add comments on the factor which is used when the statement "Good value of shape parameter ε_CA is equal to the ε for which the oscillatory behaviour begins" is presented. What is the condition here? e.g. Fourier harmonics amplitude should be larger that some threshold value etc. Only the condition algorith presented.
4) What is the rules to define the number of collocation points?
5) The review of known modifications of Fasshauer’s algorithm should be added and the benfits of the modification proposed by the Authors should be described throught the text. It is pointed mainly only that "according to the error measures used in the work, the proposed algorithms for the considered class of problem give shape parameter values that lead to clearly better results than the classic Fasshauer’s algorithm".
At the same time Authors present a parametric study on the selection of values of certain simulation parameters - the rules for the specification of the area of the parameters where the algorithms gives the appropriate results should be specified at least qualitatively.
Author Response
First, we would like to thank the Reviewers for their contribution. Their suggestions were beneficial during the revision process and have been incorporated into the revised manuscript. They helped us understand the paper's weak points and, as a direct consequence, we believe the quality of the revised paper has been improved significantly. All questions and comments are addressed below. The red fonts indicate the places of changes in the paper.
- The Abstact should be focused at the scope of Materias. Now it is very labyrinthine with numerical methods. Authors should rewrite the Abstract to emphasize the main findings of their work and convince the reader that this approach is eligible for Material science.
The abstract has been improved to better fit the scope of Materials.
- The Introduction focused mainly on the solving of heat flow equations in metamaterials. Since the author cite the Ref[1]-[4] describing the thermal metamaterials, the radiofrequency metamaterials should also be mentioned as a good candidate for the Kansa method of numerical simulation. Thus, it will be more instructive if
[Journal of Communications Technology and Electronics, 2014, Vol. 59, No. 9, pp. 914–919], [JETP Letters, 2018, Vol. 107, No. 1, pp. 25–29] are contained in the reference list along with Ref. 1-4 pointing that the Microwave metamaterials based on the reconfigurable structures can also be candidate to the beforementioned approach.
The suggested citation has been added, and the note about microwave metamaterials has been introduced.
- The Authors should add comments on the factor which is used when the statement "Good value of shape parameter ε_CA is equal to the ε for which the oscillatory behaviour begins" is presented. What is the condition here? e.g. Fourier harmonics amplitude should be larger that some threshold value etc. Only the condition algorith presented.
In Section 2.2 we have formulated the proposal of finding the good value of the shape parameter. The proposal is based on observing the main matrix condition number curve as a function of the shape parameter. The abrupt change in the regime observed on the curve graph is identified using monitoring of the curve monotonicity, which is explained in the pseudo-code Algorithm 1. The presented approach tracks the change in the sign of the observable in the moving window. If there are two monotonicity changes in a row, then it is assumed that oscillations begin there, and the shape parameter good value is found. The proposal does not require nor the Fourier transform of the observed signal nor the threshold value.
- What is the rules to define the number of collocation points?
That is an excellent question for meshless numerical methods for solving partial differential equations. In our case, the number of collocation points is selected taking into account the following issues:
- it must be large enough to achieve a sufficient spatial resolution of the solution, especially in the case of the spatially-dependent thermophysical parameters with, e.g. harmonic changes; the mentioned spatial dependence must be shown by the collocation scheme taking into account the Shannon sampling theorem;
- and at the same time it should be not too large to decrease the computational complexity of the algorithm.
It should also be emphasized that the number of collocation points is the primary parameter of the Kansa method. Using the algorithm for finding the good value of the shape parameter, we operate with the known number of the collocation points chosen at the beginning of the calculations. Then the good value of the shape parameters is identified for the assumed number of collocation points. It could be carefully assumed that the value of the shape parameter in the Kansa method is more important than the number of collocation points.
- The review of known modifications of Fasshauer's algorithm should be added and the benfits of the modification proposed by the Authors should be described throught the text. It is pointed mainly only that "according to the error measures used in the work, the proposed algorithms for the considered class of problem give shape parameter values that lead to clearly better results than the classic Fasshauer's algorithm".
At the same time Authors present a parametric study on the selection of values of certain simulation parameters - the rules for the specification of the area of the parameters where the algorithms gives the appropriate results should be specified at least qualitatively.
In the Introduction we have been present the literature review of algorithms for finding the good value of the shape parameter. We mentioned works of Hardy, Franke, Carlson and Foley, Rippa, Fasshauer and Zhang, Haung et al., Tsai et al., Esmaeilbeigi and Hosseini, Iurlaro et al. and Chen et al. According to the best authors' knowledge, no straight modifications of the Fasshauer algorithm have been proposed so far. The modification of the Fasshauer algorithm is presented in Section 2.2 and the pseudo-code Algorithm 2.
Our modification of the Fasshauer method, as for our method related to the condition number graph, explores the features of oscillatory behaviour on the cost function curve. Fig. 2 shows that the modification changes the so-called optimal or good values of the shape parameter. In fact, the presented numerical results (in Fig. 12-15) show that the modified Fasshauer algorithm gives, in most of the cases, better results than the original one, but this is not a general statement; this is only the observation based on the performed numerical experiments and introduced measures of errors.
Our parametric studies were performed additionally for the specific class of simulation parameters. We are convinced that the key in applying the Kansa method is the correct selection of the shape parameter of the radial basis functions. The additional results present only a few important cases from the point of view of the field of application, that is:
- modelling discontinuous distributions of thermal parameters, which in our cases does not require the subdomain approach,
- the influence of the Crank-Nicolson scheme parameter.
In the conclusion section, the general recommendations are given, e.g.:
- modelling discontinuous distributions of thermal parameters requires special attention because the presented results are preliminary; the various type of the Heaviside distribution approximation should be checked – może to można dopisać
- the preferred value of the Crank-Nicolson scheme parameter is 0, but the study requires an extension to the various integration schemes.